# Basement membrane diversification relies on two competitive secretory routes defined by Rab10 and Rab8 and modulated by dystrophin and the exocyst complex

Cynthia Dennis, Pierre Pouchin, Graziella Richard, Vincent Mirouse * 

Université Clermont Auvergne, Institute of Genetics, Reproduction and Development (iGReD), UMR CNRS 6293—INSERM U1103, Faculté de Médecine, Clermont-Ferrand, France

* vincent.mirouse@uca.fr

**Data Availability Statement:** All data are in the manuscript and/or supporting information files.

## Abstract

The basement membrane (BM) is an essential structural element of tissues, and its diversification participates in organ morphogenesis. However, the traffic routes associated with BM formation and the mechanistic modulations explaining its diversification are still poorly understood. *Drosophila melanogaster* follicular epithelium relies on a BM composed of oriented BM fibrils and a more homogenous matrix. Here, we determined the specific molecular identity and cell exit sites of BM protein secretory routes. First, we found that Rab10 and Rab8 define two parallel routes for BM protein secretion. When both routes were abolished, BM production was fully blocked; however, genetic interactions revealed that these two routes competed. Rab10 promoted lateral and planar-polarized secretion, whereas Rab8 promoted basal secretion, leading to the formation of BM fibrils and homogenous BM, respectively. We also found that the dystrophin-associated protein complex (DAPC) and Rab10 were both present in a planar-polarized tubular compartment containing BM proteins. DAPC was essential for fibril formation and sufficient to reorient secretion towards the Rab10 route. Moreover, we identified a dual function for the exocyst complex in this context. First, the Exo70 subunit directly interacted with dystrophin to limit its planar polarization. Second, the exocyst complex was also required for the Rab8 route. Altogether, these results highlight important mechanistic aspects of BM protein secretion and illustrate how BM diversity can emerge from the spatial control of distinct traffic routes.

## Author summary

Diversification of the basement membrane (BM) is crucial for tissue structure and organ development, yet the mechanisms governing its formation and diversification remain unclear. We focused on the *Drosophila melanogaster* follicular epithelium, characterizing BM protein secretion pathways. We identified Rab10 and Rab8 as key regulators, orchestrating parallel secretion routes. Rab10 promotes lateral and planar-polarized secretion, while Rab8 facilitates basal secretion. As a result of this highly polarized system, Rab10

**Funding:** This work was supported by the Association Française contre les Myopathies (AFM) (Trampoline grant 17683 and MyoNeurAlp network) (to VM and CD). This research was also financed by the French government IDEX-ISITE initiative 16-IDEX-0001 (CAP 20-25) (to VM and CD). The funders had no role in study design, data collection and analysis, decision to publish, or preparation of the manuscript.

route induces the formation of BM fibrils while Rab8 route generate homogenous BM. Additionally, dystrophin-associated protein complex (DAPC) is involved in fibril formation and secretion route orientation towards Rab10. Furthermore, we discovered the involvement of the exocyst complex, notably its Exo70 subunit that it interacts with dystrophin, impacting its planar polarization, and that is crucial for the Rab8 route. These findings deepen our understanding of BM protein secretion mechanisms, emphasizing the role of distinct traffic routes in BM diversity emergence. Moreover, it reveals new functions and molecular interactions for the DAPC that could be relevant for the many congenital diseases involving this complex.

## Introduction

Basement membrane (BM) is a specialized extracellular matrix forming a complex sheet-like architectural meshwork that lines most of animal tissues, such as epithelium and muscle [1,2]. BM is critical for tissue development, homeostasis and regeneration, as exemplified in humans by its implication in many congenital and chronic disorders [3]. Therefore, understanding its properties is a major issue for regenerative medicine [4,5]. BM is assembled from core components conserved throughout evolution: type IV collagen (Col IV), the heparan-sulfate proteoglycan perlecan, and the glycoproteins laminin and nidogen [2]. During development, the dynamic interplay between cells and BM participates in sculpting organs and maintaining their shape [1,6–8]. Specifically, BM influences essential cellular processes (e.g. cell identity, growth, shape, and migration) through various cell surface receptors, such as integrins [5–7,9–11]. Reciprocally, cells constantly produce and remodel their BM, thus modifying their mechanical and biochemical microenvironment [4,11,12]. BM protein secretion may involve specific proteins, mainly because of the large size of the protein complexes (e.g. procollagen) that must transit from the endoplasmic reticulum to the cell surface [13,14]. This transport is well understood from the endoplasmic reticulum to the Golgi apparatus [15,16]. However, much less is known on how these BM proteins reach the cell surface and how these routes are spatiotemporally integrated in a developing tissue.

Our understanding of BM biology has been broadened by studies in invertebrates that revealed, using fluorescent tagged versions of BM components, the unexpected diversity of BM structures and dynamics [8,17,18]. Particularly, *Drosophila melanogaster* oogenesis is a highly tractable system suitable to study BM biology. The fly ovary consists of several ovarioles, each of them composed of a succession of follicles (or egg chambers) that develop up to a mature egg. This process is categorized into 14 morphologically distinct stages [19]. Each follicle comprises a germline cyst surrounded by a monolayer of somatic epithelial follicle cells. This epithelium is polarized: the apical surface faces the germline cells, whereas the basal surface is in contact with the BM that outlines each follicle (Fig 1A). All major components of this BM are synthesized and secreted by follicle cells [20]. Several factors implicated in this process have been identified. Loss of function of the small GTPases Rab10 and Rab8, or of their respective guanine nucleotide exchange factors (GEF; Crag and Stratum) leads to the mistargeting of a fraction of BM proteins towards the apical domain. This indicates that these classical cell trafficking factors are involved in BM assembly [21–23]. However, it is not known whether these factors act together or in parallel.

Importantly, the follicle BM is implicated in egg morphogenesis [24,25]. During ovarian development, the follicle progressively elongates the initial spherical shape along the antero-posterior (AP) axis from stage 3 to adopt an ovoid shape. Elongation relies on the

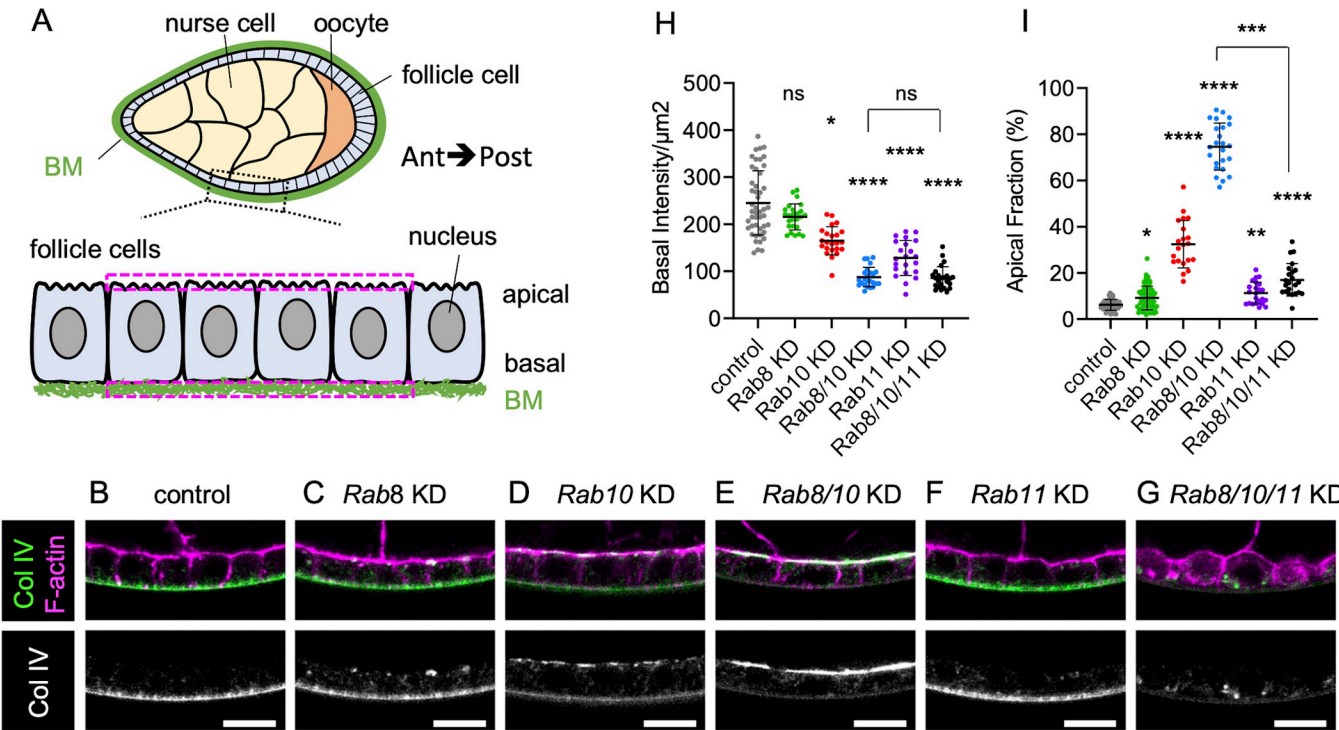

**Fig 1. Rab8 and Rab10 are both required for BM assembly.** (A) Scheme of a sagittal view of stage 8 ovarian follicle revealing inner germinal cells (nurse cells and oocyte), outer follicle cells, and basement membrane (BM) surrounding the whole follicle; zoom of the boxed region highlights the apical-basal axis of the follicle cell epithelium (Ant, Anterior; Post, posterior). This scheme and all images are oriented according to the anteroposterior axis. The two dotted magenta rectangles indicate the location of a pair of ROIs drawn on the apical and basal domains for fluorescence mean intensity measurement. (B-G) Cross-sections of stage 8 ovarian follicles showing Col IV-GFP (green, top; white, bottom) and F-actin (magenta) expression/localization in the following genotypes: (B) *tj:Gal4* (control), (C) *tj>Rab8* RNAi (Rab8 knock-down (KD)), (D) *tj>Rab10* KD, (E) *tj>Rab8>Rab10* double KD, (F) *tj>Rab11* KD, (G) *tj>Rab8>Rab10>Rab11* triple KD. Scale bar, 10 μm. (H) Measure of Col IV-GFP fluorescence basal intensity/μm$^2$ plotted for stage 8 follicles of the indicated genotypes (one-way ANOVA with Kruskal-Wallis comparisons test, n = 47, 25, 23, 24, 22, 24 pairs of ROIs). (I) Apical fraction quantification (%) of Col IV-GFP fluorescence intensity in stage 8 follicles of the indicated genotypes. (uncorrected Dunn's test; in order on graph, n = 49, 62, 21, 24, 23, 23 follicles). For all graphs, data are the mean ± SD; *p <0.05, **p <0.01, ****p <0.0001.

establishment of planar cell polarity that is compulsory to launch a process of oriented collective cell migration, perpendicularly to the AP axis, from stage 1 to stage 8 of oogenesis [24]. This migration causes the entire egg chamber to rotate inside the stationary extracellular matrix. This rotation allows the oriented deposition of BM fibrils, containing all BM components, onto a uniform unpolarized BM [24,25] (Fig 2A). It was initially proposed that BM fibrils serve as a molecular corset to mechanically constrain the follicle and direct the elongation in the AP axis. However, it has been suggested that BM fibrils act as a cue for actin fiber orientation at the basal domain of follicle cells that would be the main mechanical effector of elongation [26]. Nonetheless, BM fibrils nicely illustrate how BM diversification can elicit a morphogenetic mechanism. BM fibrils are generated from newly synthetized BM proteins that accumulate in the lateral pericellular spaces between follicular cells before being inserted in the BM into an oriented manner by the directed egg chamber rotation [25]. Rab10 overexpression leads to an increased lateral accumulation of BM proteins and of BM fibrils, suggesting its implication in the formation of these fibrils [25]. BM fibril formation requires dystroglycan (Dg), a membrane receptor for BM proteins, and dystrophin (Dys), an F-actin interacting protein [26]. Dg and Dys are the two core components of the evolutionarily conserved dystrophin-associated protein complex (DAPC) [27]. In mutant *Dys* or *Dg* follicles, the BM is devoid of BM fibrils. Moreover, epistasis experiments with Rab10 suggest a functional link between

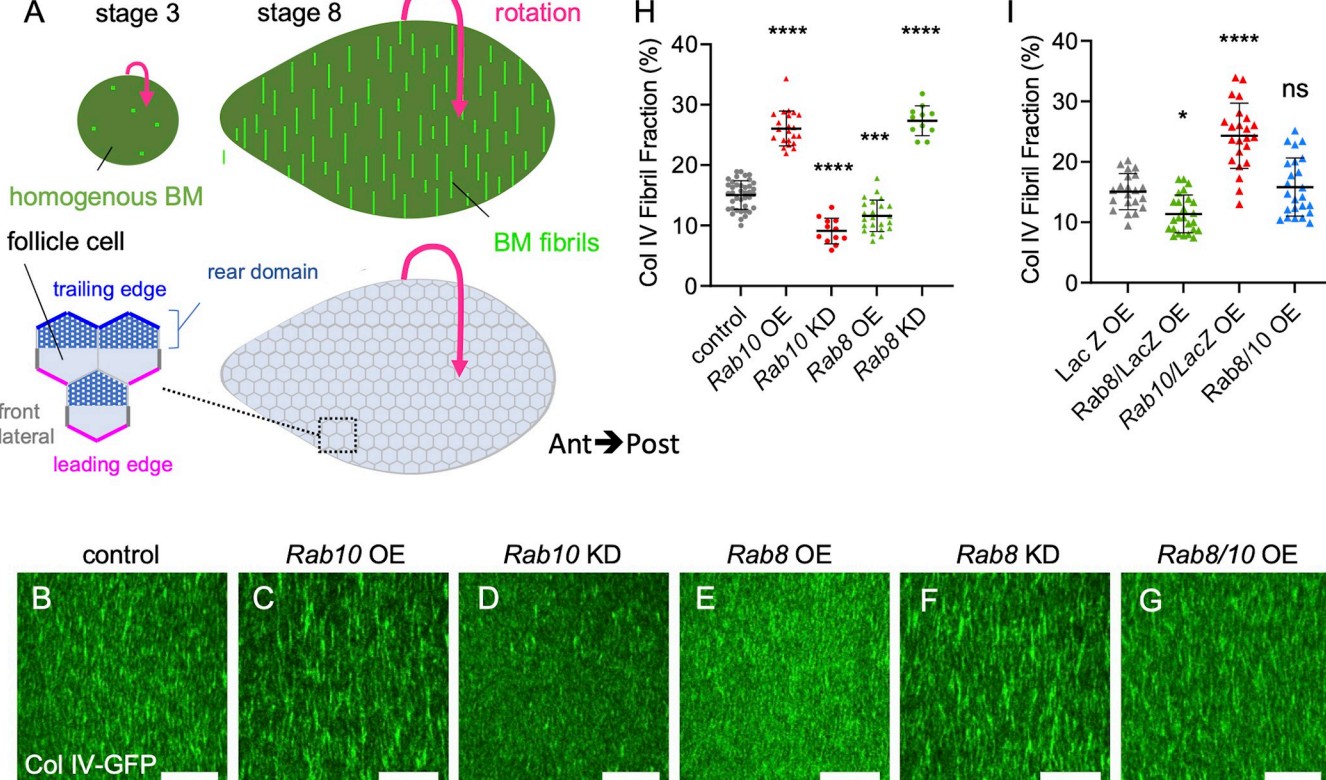

**Fig 2. *Rab8 and Rab10 compete to define two BM types.*** (A) Schemes of basal view of stage 3 (left) and stage 8 (right) ovarian follicles showing the corresponding BM type (top) and the basal surface of follicle cells (bottom); zoom of the boxed region illustrates the front (magenta) and trailing (blue) edges of follicle cells, their front lateral sides (thick grey lines) and the rear domain of the cells (dotted area) according to the orientation of the collective cell migration (indicated by the pink arrow). (B-G) Basal view of the BM (Col IV-GFP*)* at stage 8 in (B) *tj:Gal4*, (control), (C) *tj>Rab10-RFP (Rab10 overexpression (OE))*, (D) *tj>Rab10* RNAi (knock-down (KD)), (E) *tj>Rab8-YFP*, (F) *tj> Rab8* KD, (G) double *tj, >Rab10-RFP>Rab8-YFP*. Scale bars, 10 μm. (H) Quantifications of BM fibril fraction (%) in stage 8 follicles of the indicated genotypes (n = 32, 22, 12, 23, 11 follicles). (I) Quantifications of BM fibril fraction (%) in stage 8 *Col IV-GFP*-expressing follicles of the following genotypes: *tj >LacZ OE, tj>LacZ>Rab8-YFP* double *OE, tj >LacZ>Rab10-RFP* double *OE, tj>Rab8-YFP>Rab10-RFP double OE*. Data are the mean ± SD. *p <0.05, ***p <0.01, ****p <0.0001 (Ordinary one-way ANOVA with Dunnett's comparisons test). In order on the graph, n = 21, 25, 23, 24 follicles.

these actors [26]. However, how DAPC participates in BM fibril formation remains to be elucidated.

In this study, we analyzed the routes leading to BM formation in the follicular epithelium. Genetic evidence indicates that Rab8 and Rab10 define two parallel routes, forming homogenous BM and BM fibrils, respectively. Analyses of BM protein secretion in single cells revealed that this diversification is due to Rab10-driven lateral and planar polarized secretion, while Rab8 promotes basal secretion. We also found that the DAPC is part of the same cell compartment as Rab10 and influences the planar polarization of Rab10-dependent secretion. Moreover, we found that the exocyst complex is required the formation of homogenous BM.

## Results

### Rab8 and Rab10 are both required for BM assembly

To understand how BM is produced by the follicular epithelium, we first tested the role of the two small GTPases Rab8 and Rab10 by knocking them down (KD) using UAS RNAi transgenes induced by the follicle cell-specific driver tj:Gal4. BM assembly and secretion can be easily visualized at stage 8 of oogenesis using an endogenous GFP-tagged Col IV α2 chain, called

Viking. *Rab8* or *Rab10* KD induced the mistargeting of part of BM proteins toward the apical domain (Fig 1B and 1D). Similar results were obtained with three independent lines for *Rab10* and were consistent with previous observations using a transgene encoding a dominant negative form (S1A and S1B Fig) [23]. Similarly, Rab8 KD effect is identical to published data using RNAi, dominant negative form or classical mutation in mosaic clones [22]. Nonetheless, quantification of basal domain Col IV-GFP fluorescence or of fluorescence apical pool, relative to the total secreted proteins (i.e. basal plus apical pools), indicated that in both KD most proteins were still delivered basally (Fig 1H and 1I). *Rab8* KD had a weaker effect than *Rab10* KD, as indicated by the lower increase in the apical fraction compared with control. At this stage, it was unclear whether this was due to Rab8 weaker implication in BM protein secretion, redundancy with some other factor(s), or weaker *Rab8* RNAi efficiency. One attractive hypothesis was that Rab8 and Rab10, which are Rab GTPases with very similar sequences, may act in parallel. In agreement with this hypothesis, the double *Rab8-Rab10* KD completely impaired basal secretion and almost all the Col IV-GFP signal was at the apical surface, indicating that this condition is more than the simple addition of the single KD (Fig 1E, 1H and 1I). This synergic effect of *Rab8* and *Rab10* KD suggests that these two Rabs are part of alternative routes with a partial redundancy for addressing BM proteins. Moreover, the longitudinal view of the epithelium suggested the presence of a proper basement membrane, but on the apical side. However, the planar view indicated that apically secreted BM proteins assembled in aberrant discontinuous structures, as observed in other genotypes leading to apical BM secretion (S1C Fig) [28].

It has been previously shown that the apical secretion observed in the *Rab10* KD can be suppressed by *Rab11* KD, suggesting that Rab11 may define a cryptic route for BM protein secretion [23]. We tested this hypothesis by knocking down all three genes (*Rab8*, *Rab10* and *Rab11* triple KD). In this condition, apical secretion was completely blocked, and BM proteins were in big intracellular blobs (Fig 1G and 1I). Notably, basal secretion was not restored, indicating that Rab11 did not compete with any other route besides Rab8 and Rab10 (Fig 1G and 1H). Altogether, these data indicate that Rab8 and Rab10 define the proper path for BM secretion in a partially redundant manner, and that no other route towards the basolateral side of the follicular epithelium exists.

Considering the important role of BM in epithelial architecture, we wondered whether completely reversing BM secretion towards the apical domain (as observed in the *Rab8-Rab10* double KD) or lack of BM secretion (*Rab8*, *Rab10* and *Rab11* triple KD) could affect epithelial polarity. Therefore, we checked the localization of atypical protein kinase C (aPKC), an essential actor of the apical domain definition commonly used as a marker of this domain, Discs large protein (Dlg), known to be located at the lateral cortex, and E-cadherin protein (Ecad) accumulated at the zonula adherens [29]. We observed no difference between control and *Rab8-Rab10* double KD (S1D Fig). Similarly, in the *Rab8*, *Rab10*, *Rab11* triple KD, aPKC and E-cad reduction respectively at the apical surface and the zonula adherens were not stronger than in the single *Rab11* KD, indicating that the defect observed in these contexts was independent of BM secretion impairment (S1D Fig). These data show that in the follicular epithelium, BM localization/presence has no major impact on polarity maintenance. These observations are in agreement with previous data indicating that major actors of the cell-extracellular matrix interface, such as integrins and Dg, are not required for cell polarity in the follicular epithelium [30].

## Rab8 and Rab10 compete to define different BM types

Follicle BM is composed of an homogenous matrix from the very first stages while BM fibrils are added during the collective cell migration (Fig 2A, top) [24,25]. Although the exact

contribution of each of these BM types is not yet fully understood, genetic manipulation indicated that they are both required for the proper morphogenesis of the future egg [24–26]. Findings mainly based on gain of function experiments suggest that Rab10 participates in the follicle cell BM diversification by contributing to the formation of BM fibrils that are deposited as the cells migrate [25]. On the other hand, the route to generate homogenous BM remains unknown. Therefore, we analyzed the effects of Rab8 or Rab10 KD or overexpression (OE) on BM structure by assessing Col IV-GFP pattern at stage 8 (i.e. the end of BM fibril deposition). We quantified the fibril fraction (FF) that corresponds to the percentage of fluorescence detected in fibrils relative to the total BM fluorescence at the follicle basal surface. As previously observed, *Rab10* OE led to FF increase (Fig 2B, 2C and 2H). Conversely, *Rab10* KD significantly decreased FF, providing the first loss of function evidence of its involvement in BM fibril formation (Fig 2D and 2H). Conversely, Rab8 OE decreased the FF, whereas *Rab8* KD increased it (Fig 2E, 2F and 2H). Thus, Rab8 and Rab10 had exact contrary effects. Importantly, as in *Rab8* KD part of BM proteins was targeted to the apical surface and as the FF was relative to the total BM fluorescence at the follicle basal surface, its increase in *Rab8* KD might partly reflect a decrease of the homogenous matrix. Thus, Rab8 defines a different route than Rab10, leading ultimately to a different BM type. Previous observations indicated that a strong increase in BM fibril formation might induce a depletion of the homogenous matrix, suggesting a competition between the mechanisms promoting the two BM types [25]. We tested this hypothesis by comparing the concomitant and the single overexpression of *Rab8* and *Rab10* (combined with a *UAS:LacZ* transgene as a control for a potential Gal4 titration). The concomitant Rab8-Rab10 OE led to a FF value similar to control (Fig 2G and 2I). Altogether, these results show that Rab8 and Rab10 define two distinct routes, each responsible for a different BM type (homogenous matrix and fibrils, respectively) and in competition with each other.

## Rab10 orients BM secretion towards the lateral rear cortical domain

To explain how Rab8 and Rab10 allow the formation of distinct BM types, we investigated the cell exit site of BM proteins. It has already been shown that Rab10 OE increases the lateral secretion of BM proteins, using immunostaining against BM proteins without permeabilization [25]. However, this approach does not precisely map the exit site in single cells because it does not allow determining BM protein origin between cells and detecting any potential basal secretion. Therefore, we combined immunostaining without permeabilization with the M-TRAIL tool, initially developed to follow the migration of individual follicle cells *in vivo* [31]. M-TRAIL generates single-cell clones that express a Col IV subunit protein fused to GFP, named Cg25c (UAS:Col IV-GFP). Each cell forms a fluorescent trail that corresponds to its collagen secretion while migrating (Fig 3A and 3B). Thus, detection of Col IV with an anti-GFP antibody and a Cy3- or Cy5-conjugated secondary antibody without permeabilization allowed discriminating secreted collagen from the total protein. First, we verified that the expression of the UAS:Col IV-GFP transgene in the whole tissue using the follicle cell-specific driver tj:Gal4 led to the formation of a fluorescent BM, like the one observed with endogenous Col IV-GFP, indicating that the transgene was normally incorporated in the different BM types (S2 Fig). Then, we analyzed Col IV-GFP secretion in single cells.

   In controls (wild type), we observed that the lateral secretion occurred mostly close to the basal surface, usually at a restricted area of <1 μm high that we named suprabasal region (Fig 3C'). We could also detect some basal secretion, with foci of BM proteins at cell surface (Fig 3C"). These foci were small and did not assemble in fibrils. Importantly, since we did detect almost no signal in the lateral front region, these foci were unlikely to be due to lateral secretion that reached the basal surface and were beneath the cell while migrating. Moreover, we

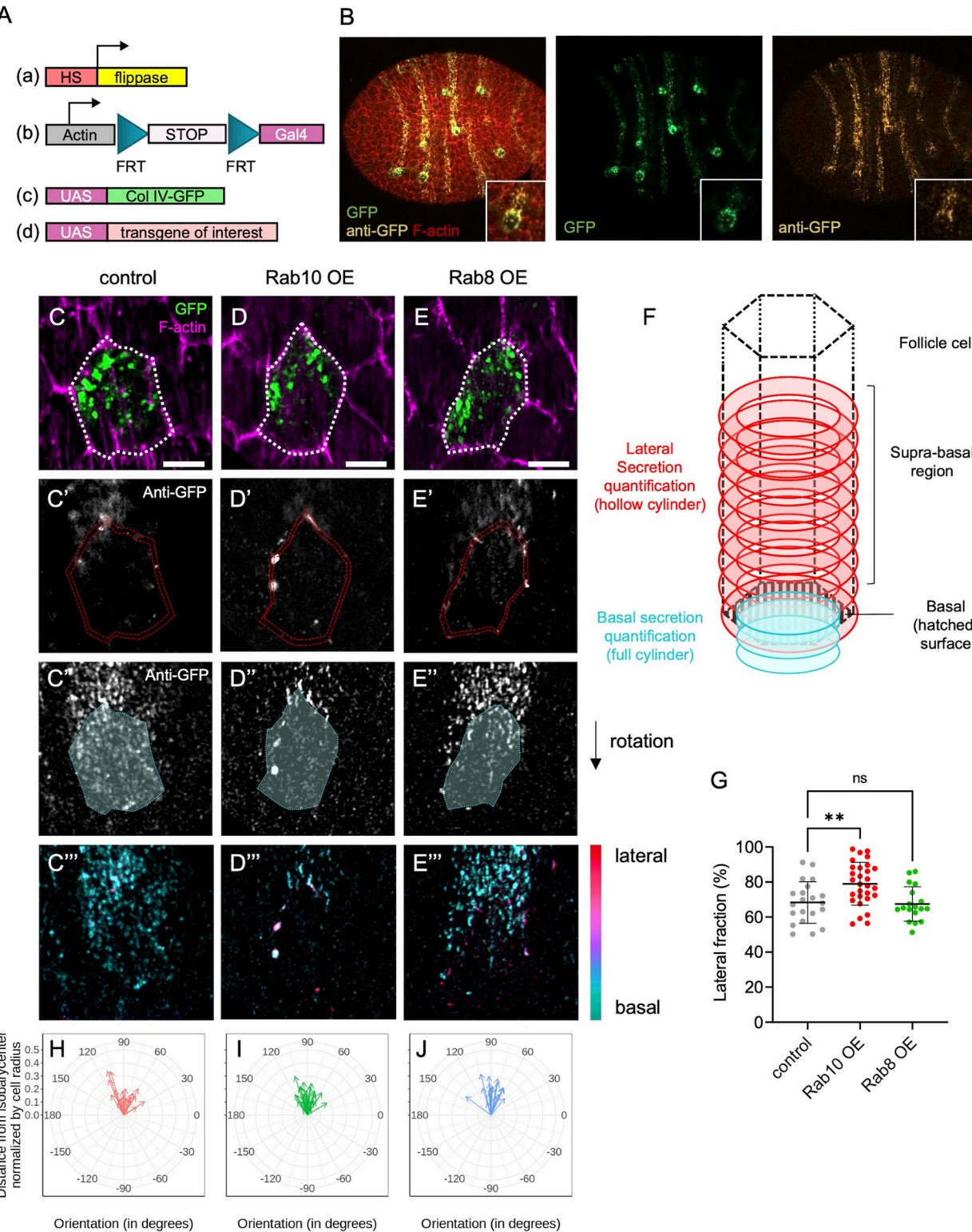

**Fig 3. *BM proteins are secreted from two distinct subcellular domains.*** (A) Scheme of the genetic components used to generate single-cell clones that overexpress Col IV-GFP and the protein of interest. (B) Image of a whole stage 8 follicle after induction of Col IV-GFP expression (green) in single-cell clones stained for F-actin (red) and immunostained for GFP (yellow) without permeabilization to label secreted Col-IV-GFP. (C-E) lateral (C-E') and basal (C"-E") focal planes of individual single-cell clones of the indicated genotypes expressing Col IV-GFP (green, top), stained for F-actin (magenta) and immunostained for GFP without permeabilization (white). Dotty white line (C-E) highlights the cell periphery. Double

dotty red lines (C'-E') and white surface delimited by dotty cyan line (C"-E") indicate the surface of the given z-section considered for lateral or basal secretion quantification respectively. Color-coded z projection in function of the z axis of the lateral and basal 12 focal planes used for BM secretion quantifications (C"'-E"'). Scale bars, 5 μm. (F) Scheme of how BM secretion was quantified: lateral secretion was assessed using a hollow cylinder (red) of 1.5μm thickness based on cell cortex segmentation. Basal secretion was assessed using a full cylinder (cyan) made by projecting the ROI drawn on the basal surface and shrunk to 0.75μm in diameter on the three slices beyond the cell surface. (G) Quantification of the lateral fraction of BM secretion (%) in single-cell clones of the indicated genotypes overexpressing Col IV-GFP. In order on the graph, n = 21, 29, and 18 cells. Data are the mean ± SD; **p <0.01 (Ordinary one-way ANOVA with Dunnett's comparison test). (H-J) Quantification of the planar orientation of lateral BM protein secretion from individual single-cell clones of the following genotypes: control (red), Rab10 OE (green), Rab8 OE (blue). Each arrow represents the vector between the isobarycenter of the hollow cylinder and its center of mass for one cell, using the signal intensity as weight. Images were oriented in such a way that the cell trailing edge points toward 90˚.

usually observed these foci on the rear half of the cell (Fig 3C"'). We developed a method to quantify these observations. Briefly, we segmented cells in 3D according to their F-actin signal. Then, we separately quantified the GFP signal at the cell cortex and basal cell surface, allowing us to determine the lateral fraction (Fig 3F). This showed that in controls, ~70% of the BM was laterally secreted (Fig 3G). We also observed that this lateral secretion was highly planar-polarized at the rear of cells (Fig 3C'). We could estimate the planar polarization by calculating a vector that corresponded to the distance and orientation between the barycenter of a cylinder representing the cell wall and the lateral signal. This confirmed the robust planar polarization of the lateral secretion (Fig 3H). Altogether, these data indicated two major sites of BM secretion: one basally, and one laterally. Lateral BM secretion was predominant and highly planar-polarized.

Importantly, our approach to study BM secretion in single cells can be associated with additional UAS transgenes to overexpress genes of interest and to visualize their cell autonomous impact. This allowed us to show that *Rab10* OE in single cells strongly increased the polarized lateral fraction and depleted the basal secretion (Fig 3D–3D"' and 3G). *Rab10* OE did not modify the planar polarization of the basal fraction (Fig 3I). We could not detect a clear effect of *Rab8* OE (Fig 3E–3E"', 3G and 3J), possibly because this transgene had a milder effect on BM (Fig 2). Nonetheless, together with the previous results, Rab10 OE effect suggests a model in which Rab10 guides the planar polarized lateral secretion, whereas Rab8 may promote basal secretion.

## Rab10 and dystrophin associate with a tubular compartment that contains BM proteins

To understand how Rab10 contributed to the lateral planar-polarized secretion of BM proteins, we compared the subcellular localization of Rab10 and of BM components. It has already been shown that Rab10 and collagen are localized at the rear of follicle cells considering the rotation orientation [23]. Super-resolution images, obtained with an AiryScan microscope, of endogenous tagged Rab10 (Rab10-YFP) revealed a complex distribution (Fig 4A–4D). This pattern was reproduced by expression of the *UAS:Rab10-RFP* transgene that is easier to detect and at a temperature where this transgene has no gain of effect on FF [26]. Moreover, this transgene allowed the concomitant observation of Col IV-GFP (Figs 4E–4H and S3A–S3C). We observed three main localization profiles from the cell center to the basal surface. 1) At ~1–2μm from the basal cell surface, we detected big Rab10-positive vesicles, usually at the rear of cells and juxtaposed to Col IV-containing vesicles (Figs 4A, 4E and S3A). Based on published data, this localization is close to exit sites from the Golgi apparatus, as confirmed by the colocalization with Golgin-245, a marker of trans-Golgi network (S3D Fig) [23,28]. 2) We also observed Rab10 at the rear lateral cell cortex, mainly in the suprabasal region. This cortical localization overlapped with the area where lateral secretion occurred. Therefore, we assumed that it corresponded to the cell exit site of the Rab10 route (Figs 4B, 4F and S3B). 3) More

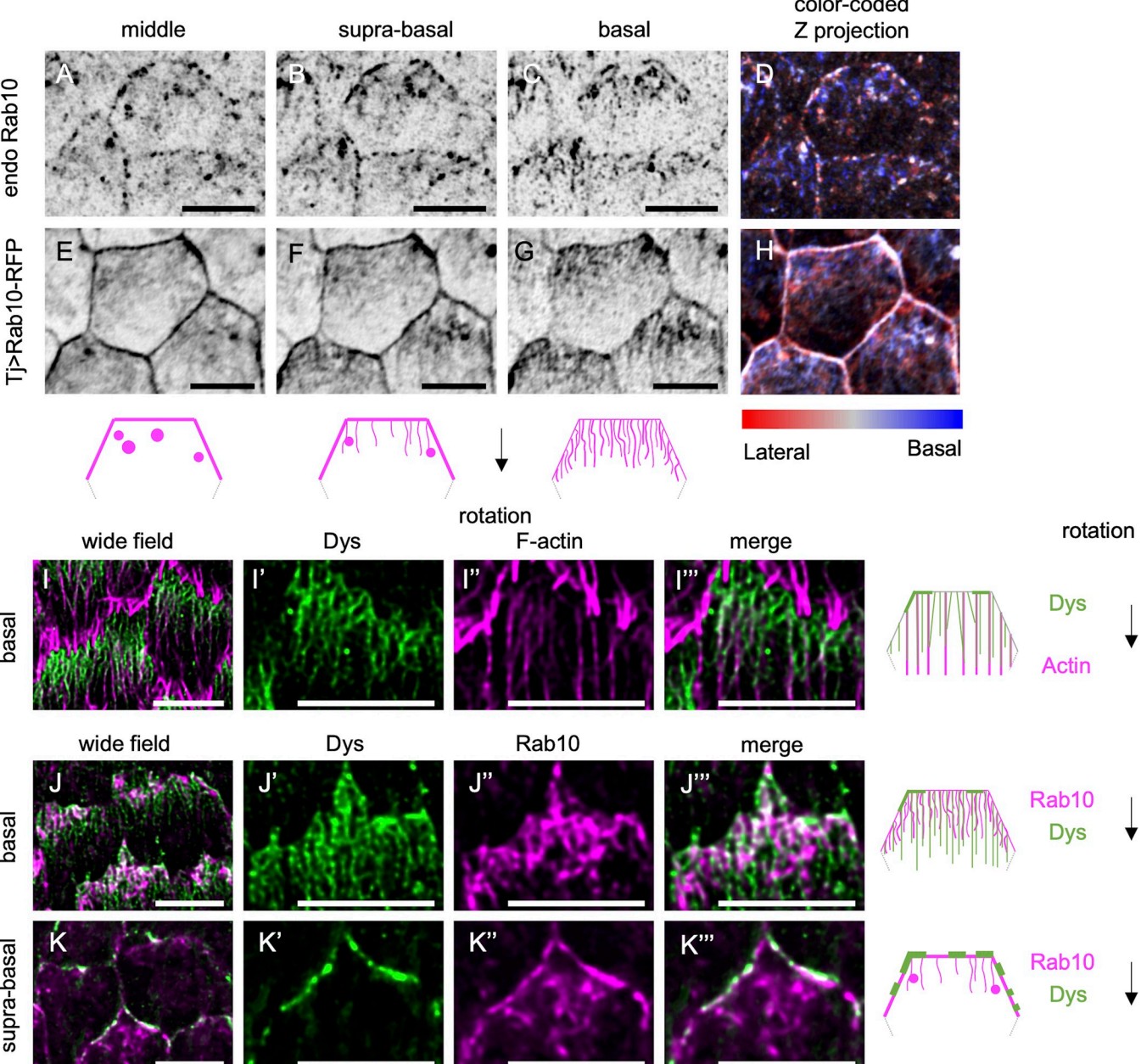

**Fig 4. *Rab10 and Dys associate with tubular structures.*** (A-H) Basal (C,G), suprabasal (i.e. 0.4 μm above basal surface) (B, F), and middle (i.e. 0.8 μm above basal surface) (A, E) views or color-coded z projections as a function of the z axis starting from the basal surface (D, H) of stage 8 follicles that express endogenous YFP-tagged *Rab10* (A-D) or *tj> Rab10-RFP* (E-H). Schemes represent the rear half of a follicle cell with the localization of Rab10 (magenta) at the basal (right), suprabasal (center), and middle cell levels (left). Black arrow indicates the direction of follicle rotation. (I) Basal view of stage 8 follicles expressing endogenous Dys-sfGFP (green). F-actin is visualized in magenta. (J, K) Basal (J-J''') or suprabasal (K-K''') view of stage 8 follicles that express endogenous *Dys-sfGFP* (green) and *tj> Rab10-RFP* (magenta). Schemes represent the rear half of a follicle cell with the localization of actin or Rab10 (magenta) and Dys (green) at the basal (top and middle) or suprabasal plane (bottom). Scale bars, 5 μm.

basally, we detected Rab10 in a tubular compartment. Rab10 association with tubular endosomes was reported in mammalian cell culture and in *Caenorhabditis elegans* [32,33]. We detected these tubule-like structures at the rear of cells. They were usually oriented in the migration axis in such way that their extremity was very close to or even reached the lateral

rear of the cells (Fig 4C and 4G). As the tubular compartment is very close to the cell basal surface (i.e. below the z resolution of AiryScan), the presence of a very strong Col IV-GFP signal coming from the BM precluded its visualization in the intracellular basal compartment (S3C Fig). To circumvent this problem, we used two strategies. First, we looked at Col IV-GFP and Rab10-RFP expression in samples treated with collagenase before fixation (S3E Fig). Second, we performed FRAP experiments. After bleaching the Col IV-GFP signal on a large area at the surface of living follicles, we waited for 20 min to allow new cells to migrate to this area, and then acquired images of Rab10-RFP and Col IV-GFP signals at the basal surface of these cells (S3F Fig). In both approaches, we observed a partial colocalization of Col IV and Rab10 in the most basal Rab10-positive compartment. As this compartment was very close to the trailing edge of the cell where BM proteins are secreted and where Rab10 is also localized, we assumed that it corresponded to an important intermediate step of this secretory route.

Dys and Dg are both required for BM fibril formation, suggesting that they may be part of the Rab10 route [26]. To better understand their contribution, we knocked in superfolder GFP (sfGFP) at the C-terminus of Dys to label all known isoforms produced by the *Dys* locus. Dys-sfGFP was enriched at the basal surface of follicular cells and its localization was strongly planar-polarized at all the stages of follicle rotation with an enrichment at the rear of cells (Fig 4I and 4J). Super-resolution images revealed that Dys drew parallel filaments at the basal domain that extended to approximatively half of the cells at stage 8 and showed a strong colocalization with actin stress fibers (Fig 4I). Dys was also localized at the lateral trailing edge in the suprabasal region (Fig 4K). It also strongly colocalized, though not perfectly, with Rab10, both at cell cortex and at the tubular compartment (Fig 4J–4K). Altogether, these data identified Dys and Rab10 as part of a highly planar polarized subcellular compartment, with structures reminiscent of tubular endosomes. This compartment contained BM proteins and was oriented towards the BM protein cell exit site, suggesting that Rab10 and the DAPC act together to regulate their lateral secretion.

## DAPC is sufficient to recruit and redirect the Rab10-dependent secretory route

To investigate DAPC involvement in the Rab10 route, we analyzed their functional relationship. We previously showed that Rab10 overexpression can rescue FF in follicle cells in which *Dys* was knocked down by RNAi, suggesting that DAPC function in BM fibrils might be accessory [26]. However, when we overexpressed *Rab10* in a *Dys* null mutant, FF was not rescued, indicating that Dys presence is obligatory for BM fibril formation (Fig 5A–5D). Nonetheless, the absence of BM protein apical secretion in the *Dys* mutants indicated that they behaved differently than a Rab10 loss of function (Fig 5G, 5H and 5J).

We hypothesized that DAPC recruits vesicles containing BM proteins for the Rab10 route and that in its absence, all BM proteins use the Rab8-dependent route, leading to the exclusive formation of homogenous BM. In this scenario, *Dys* loss should completely suppress the apical secretion observed in Rab10 KD cells and block BM formation in the absence of Rab8. As apical secretion was not suppressed in *Dys* null mutant follicles upon *Rab10* KD, and even slightly increased, we excluded this hypothesis (Fig 5G and 5I). However, apical secretion was strongly increased in *Dys* null mutant follicles upon *Rab8* KD, although it was not sufficient to block basal BM formation (Fig 5G, 5K and 5L). We interpret this result as the consequence of the competition between the Rab8 and Rab10 routes in a genetic context where the Rab10 route is affected, ultimately leading to more BM proteins transported by the Rab8 route. Moreover, in this genetic combination (*Dys* null mutant + *Rab8* KD) there is no FF rescue (Fig 5A and 5D–5F). Thus, in the absence of Dys, a significant fraction of BM proteins still used the Rab10

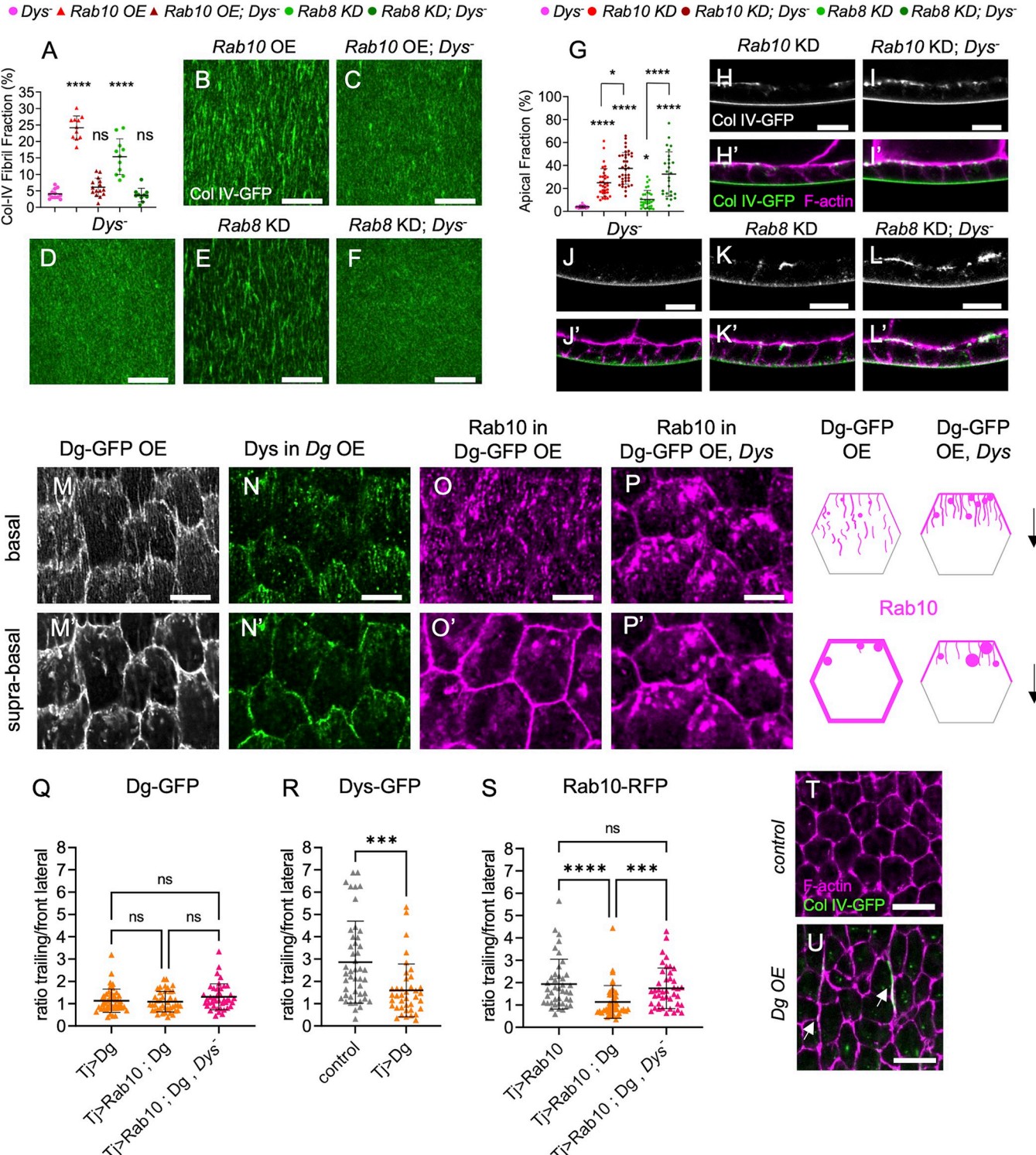

**Fig 5. DAPC is sufficient to recruit the Rab10-dependent route.** (A) Quantifications of BM fibril fraction (%) in stage 8 follicles of the indicated genotypes (n = 12, 11, 15, 11, 10 follicles) (B-F) Basal view of the BM in stage 8 *Col IV-GFP*-expressing follicles of the following genotypes (B) *tj> Rab10-RFP (Rab10* OE*)* (C) *tj >Rab10-RFP; Dys* (D) *Dys* (E) *tj,>Rab8 RNAi (Rab8 KD)*, (F) *tj, >Rab8 RNAi; Dys*. Scale bars, 10 µm. (G) Apical fraction quantification (%) of Col IV-GFP fluorescence intensity in stage 8 follicles of the indicated genotypes (In order on graph, n = 32, 34, 37, 32, 26 follicles). (H-L') Cross-sections of stage 8 *Col IV-GFP*-expressing follicles of the following genotypes: (H, H') *tj>Rab10 RNAi (Rab10 KD)*, (I, I') *tj>Rab10 RNAi; Dys*, (J, J') *Dys*, (K, K') *tj>Rab8 RNAi*, (L,L') *tj>Rab8 RNAi; Dys*. Images show Col IV-GFP (green, bottom; white, top) and F-actin (magenta). Scale bar, 10 µm. (L) Apical fraction quantification (%) of Col IV-GFP fluorescence intensity in stage 8 follicles of the indicated genotypes (In order on graph, n = 32, 34, 37, 32, 26 follicles). (M-P) Basal and (M'-P')

suprabasal views of stage 8 follicles of the following genotypes: (M, M') *tj> Dg-GFP* (DG-GFP overexpression (OE), (N, N') *tj> Dg; Dys-sfGFP*, (O, O') *tj> Dg-GFP > Rab10-RFP*, (P, P') *tj> Dg-*.5 μm. (Q-S) Quantification of (Q) Dg-GFP, (R) Dys-sfGFP and (S) Rab10-RFP mean signal intensity at the trailing edge vs adjacent lateral front side of follicle cells of the indicated genotypes, on 0.45 μm projections starting 0.15 μm below the basal surface; n = 46, 41, 42 follicle cells (Q), n = 46, 39, follicle cells (R), n = 41, 43, 42 follicle cells (S). ****p <0.0001, ***p <0.001, *p<0.05, ns: not significant (Ordinary one-way ANOVA with Tukey's multiple comparisons test for A, Kruskal-Wallis with Dunn's comparison test for Q, S, or with uncorrected Dunn's comparisons test for G, and Mann-Whitney test for R). In all graphs, data are the mean ± SD. (T-U) Representative images of F-actin (magenta) and Col IV-GFP (green) in (T) control and (U) Dg-overexpressing follicle cells in a plane in the middle of the cells. Note the presence of large BM fibrils between cells (white arrows) and the aberrant shape of cells in U. Scale bars, 10 μm.

route, but did not generate fibrils. Next, we asked whether Dys could regulate the subcellular targeting of the Rab10 route. In *Dys* null mutant clones we did not observe any change in Rab10-RFP localization compared with the neighboring wild type cells (S4A Fig). We then quantified Rab10 polarized localization at the cell cortex by calculating the mean signal intensity ratio between the boundaries corresponding to the trailing edge of the follicle cell and that corresponding to the front lateral side, parallel to the migration axis (Fig 2A, bottom). We did not find any difference (S4B Fig). We analyzed BM secretion at single-cell resolution in *Dys* null mutant cells and did not detect any significant difference in the ratio between lateral and basal secretions and in the planar polarization of the lateral secretion (S4C and S4D Fig).

To better understand the relationship between DAPC and Rab10, we also induced *Dg* OE in the follicular epithelium. Endogenous Dg was described by immunostaining to be mainly enriched on the basal side of the cells [21,34]. When overexpressed, the protein localized at the basal surface and also all along the lateral and apical cortex of follicle cells (Figs 5M and S4E). We think that this Dg localization was ectopic and due to its overexpression. Indeed, it induced the recruitment of Dys-sfGFP to the whole cell periphery where Dys is normally absent, and depleted Dys from the basal tubular compartment (Figs 5N and S4F–S4G). Consequently, Dys cortical localization was no more planar-polarized (Fig 5N' and 5R). In conditions of overexpression (29°C), Rab10 is still planar polarized (Fig 5S). Strikingly, Dg OE led to Rab10 recruitment to the whole lateral cortex, abolishing its planar polarization (Fig 5O and 5S). This suggested a physical link between the DAPC and Rab10. As Dys is required for BM fibril formation, such Rab10 recruitment upon Dg OE might be related to BM fibril formation, and is likely to be Dys-dependent. We tested this hypothesis by overexpressing *Dg* in a *Dys* null mutant background. In this context, Rab10 was not recruited to the lateral cortex and was still planar polarized, whereas Dg was still located at the whole cell periphery (Figs 5P, 5Q, 5S and S4H–S4I). Thus, Dys is required for Rab10 recruitment by Dg.

Interestingly, Rab10 delocalization induced by Dg OE allowed us to test specifically the role of the planar-polarized lateral secretion versus unpolarized secretion from the whole lateral periphery. We analyzed BM protein secretion in Dg OE follicle cells. FF quantification revealed a slight increase following Dg OE (S4J–S4L Fig). However, probably as a direct consequence of Rab10 redistribution all around the cells, we noticed the presence of very long and intense fibrils that remained trapped between the lateral domain of the follicle cells in the axis of cell migration (Figs 5U and S4L). As recently reported in another genetic context affecting BM secretion routes and where similar lateral fibrils were described, their presence disrupted the cell geometry (Fig 5T and 5U and [28]). Altogether, these data underly that BM protein secretion planar polarization is required to maintain the epithelial structure. They also show that Dg is sufficient to reorient the Rab10 route.

## Exo70 interacts with and limits dystrophin planar polarization

To understand how Rab10 and the DAPC could be connected, we noticed that the exocyst subunit Exo70 was identified as a Dys interactor using a large scale two-hybrid proteomic

approach [35]. Exocyst is an octameric complex, very conserved from yeast to humans. It is usually involved in the subcellular targeting of exocytosis by tethering vesicles to a specific location on the plasma membrane, but it can also participate in vesicle transport or docking [36,37]. A physical interaction between Dys and the exocyst complex could be likely because this complex has been functionally linked with Dys and Rab10 in other contexts [38–42]. Although yeast two-hybrid approaches may generate false positives, mining of initial data indicated that multiple overlapping fragments of Dys were identified in a screen with fly Exo70 as prey, strongly increasing the confidence (Fig 6A) [35]. Dys long isoforms include an actinin domain that can bind to F-actin, a long rod domain that contains spectrin repeats (SR), and a C-terminal domain that includes the Dg binding region. The minimal interaction domain with Exo70, deduced from the yeast two-hybrid results, overlapped with two SRs that corresponded to amino acids 2549–2635 on the longest fly Dys isoform (RH). Based on their position and homology, these two SRs correspond to SR22 and SR23 in mammalian dystrophin (Fig 6A). The interacting domain also corresponded to the most conserved part of the whole rod domain from fly to humans (53% and 70% of amino acid identity and homology, respectively). However, no clear function or protein interaction has been assigned to it yet. GST pull-down experiments with a Dys SR22-SR23-containing fragment and Exo70 confirmed their direct interaction (Fig 6B). Overexpressed Exo70 fused to the Scarlet fluorescent protein localized to the whole cell cortex (Fig 6D and 6H). Exo70 OE led to Dys recruitment (Fig 6C and 6D–6D') and reduced Dys cortical planar polarity *in vivo* (Fig 6F and 6K). However, unlike Dg OE that recruits both Dys and Rab10, Exo70 OE did not affect Rab10 polarized localization (Fig 6G, 6H–6H' and 6J). Altogether, these data indicate that Dys and Exo70 interact directly. They also suggest that this interaction is not implicated in the DAPC-Rab10 interaction, and consequently exocyst may not provide a link between them.

In *Exo70* null mutant follicles, Dys localization was hyperpolarized in the suprabasal region with a stronger signal at the trailing edge of the cells, the opposite of what observed in Exo70 OE (Fig 6E, 6F and 6K). We also observed a stronger Rab10 localization in the same domain in *Exo70* null mutants, although the effect was less obvious than for Dys (Fig 6I and 6J). Thus, Exo70 limits Dys planar polarization and consequently that of Rab10, confirming that Dys-Exo70 interaction does not provide a link between Rab10 and Dys.

To uncover a potential role of Dys-Exo70 interaction, we analyzed exocyst involvement in BM formation. In *Exo70* null mutants, which are viable, FF was strongly increased (Fig 6L, 6N and 6P). We reproduced this defect by inducing, in the follicular epithelium, RNAi against *Exo70* and against the other seven subunits of the exocyst complex (S5A Fig). Moreover, we confirmed that in the *Exo70* null mutant, the BM defect could be rescued by Exo70 OE in follicle cells (S5B–S5E Fig).

The *Exo70* null mutant impact on BM is consistent with Dys and Rab10 hyperpolarization. As exocyst and Dys loss of function show opposite effects on BM, we could perform an epistasis test. We did not manage to recover double *Dys* and *Exo70* null mutant flies, therefore we induced *Exo70* KD in follicle cells in a *Dys* null mutant background. The FF decrease to the level observed in the *Dys* mutant clearly indicated that *Dys* was epistatic to *Exo70* (Fig 6M, 6O and 6P). This finding favored a model in which Exo70 limits BM fibril formation by modulating Dys localization.

## Exocyst is required for the Rab8-dependent route

We also analyzed the effect of the *Exo70* null mutant on BM secretion at the single-cell level. Lateral secretion was still oriented towards the rear of the cells, consistently with the fact that the Dys-Rab10 route was still planar polarized (Fig 7A and 7B). Conversely, the basal secretion

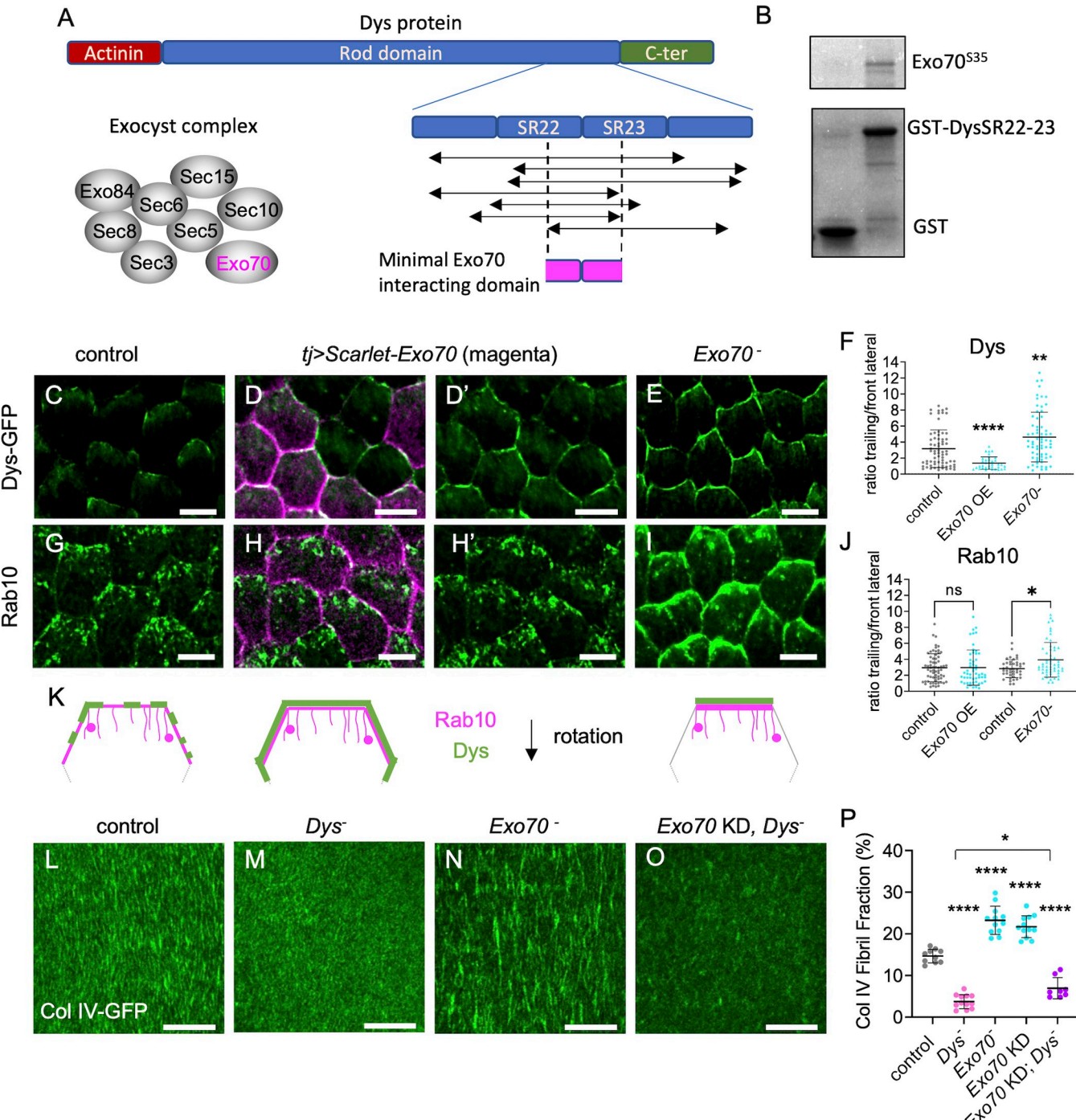

**Fig 6. *Exo70 interacts with and limits the planar polarization of dystrophin.*** (A) Representative scheme of Dys domains and the fragments identified in a yeast two-hybrid screen with Exo70 as prey [35]. These fragments encompass a minimal region that includes some spectrin repeats (SR 22 and 23). (B) GST pull-down experiments between GST (left) or GST-SR22-23 (right) and S[35]-labeled Exo70. (C-E, G-I). Suprabasal view of stage 8 follicles that express (C-E) endogenous *Dys-sfGFP* (green), (G-H) *tj>Rab10-YFP* (green) or (I) *tj>Rab10-RFP* (green), in (C, G) control, (D, D', H, H') *tj> Scarlet-Exo70* (Exo OE; magenta) or (E, I) *Exo70* null mutant. Scale bars, 5 μm. Note that Scarlet-Exo70 is expressed in a mosaic manner and that Dys lateral delocalization fits with this expression pattern. (F, J) Quantification of Dys-GFP (F) and Rab10-YFP (J, left) or Rab10-RFP (J, right) mean signal intensity at the trailing edge vs adjacent lateral front side of follicle cells of the indicated genotypes, on 0.45 μm projections starting 0.15 μm below the basal surface n = 112, 45, 89 follicle cells (F), n = 66, 54, 108, 124 follicle cells (J). (K) Schemes representing the rear half of follicle cells with the localization of Dys (green) and Rab10 (magenta) at the suprabasal surface in control (left), *Exo70* OE (middle) and *Exo70* null mutant (right). (L-O) Basal view of the BM at stage 8 in (L) control, (M) *Dys* null mutant, (N) *Exo70* null mutant, (O) double *tj>Exo70* RNAi (Exo KD)-*Dys* null mutant follicles visualized with Col IV-GFP (green). Scale bars, 10 μm. (P)

Quantification of BM fibril fraction (%) in stage 8 follicles of the indicated genotypes. In order on the graph, n = 10, 12, 12, 12, 8 follicles. Data are the mean ± SD; * p <0.05, ** p <0.01, **** p <0.0001 (Kruskal-Wallis test with uncorrected Dunn's comparisons test for F and J, and with Tukey's multiple comparisons test for P).

was strongly impaired with almost no BM protein detected at the basal surface of Col IV-GFP expressing cells (Fig 7A and 7C). Therefore, we asked whether Exo70 may interfere with BM formation by an additional way than Dys hyperpolarization, and whether this could influence the Rab8 route. Three hypotheses could explain the *Exo70* phenotype: 1) the Rab8 route is mistargeted towards the lateral domain to make BM fibrils like in the Rab10 route; 2) all BM proteins are directed towards the Rab10 route; 3) the Rab8 route is blocked and cannot induce basal secretion. If the Rab8 route were mistargeted, then Rab8 loss, which normally increases FF, should decrease FF in the *Exo70* loss of function context. However, we did not observe any FF change in WT and *Exo70* KD follicles upon *Rab8* KD. This ruled out the first hypothesis (Fig 7D). If all BM proteins were taken in charge by the Rab10 route in the *Exo70* null mutant, then Rab10 overexpression should not enhance the impact on FF. However, Rab10 OE in the *Exo70* null mutant background strongly increased FF, which tends to exclude the second hypothesis (Fig 7D).

If the Rab8 route were blocked, then concomitantly blocking the Rab10 route should completely inhibit basolateral secretion. Strikingly, in the *Rab10* and *Exo70* double KD, we could not detect Col IV-GFP at the basal surface of the follicular epithelium (Fig 7E–7H). However, in such condition there is almost no apical secretion, compared for instance to the *Rab10* and *Rab8* double KD (Fig 1), suggesting a complete inhibition of the routes for BM formation. Also, we never observed apical secretion in *Exo70* KD follicles (Fig 7E), indicating that *Exo70* KD effect was different compared with that of *Rab8* KD. Together, these results suggest that Exo70 is required for the Rab8-dependent BM route and at least to some extent, for the Rab11 cryptic route. Importantly, they also strongly argue for a basal targeting of the Rab8 route.

Lastly, we checked whether Exo70 effect on the Rab8 route was independent of its effect on Dys localization. As previously shown, in the *Rab10-Exo70* double KD, there was no basolateral secretion, excluding the hypothesis that Exo70 acts exclusively through hyperpolarization of the Rab10 route (Fig 7G). Conversely, *Rab8 KD* slightly affected Dys localization, but, importantly, this effect is opposite to the one observed in the *Exo70* null mutant (Fig 7I). These results tend to confirm that Exo70 participates in follicle BM secretion via two independent mechanisms.

## Discussion

Proper BM protein secretion and BM diversification are essential for development. Our work brings new mechanistic insights into how BM proteins are secreted in epithelial cells and demonstrates that BM diversification during development can arise from the existence of different secretory routes with different cell exit sites.

Strikingly, these routes are highly integrated in the different polarity axes (i.e. apical basal cell polarity and planar cell polarity) (Fig 7J). First, we refine the idea that Rab10 route is laterally targeted by showing that it is planar-polarized towards the trailing edge of the cells. Second, our results provide indirect but strong evidence that Rab8 targets a route towards the basal domain. While Rab8 and Rab10 are in competition, basal secretion is depleted by Rab10 OE. Moreover, all the results from the analysis of *Exo70* mutants converge on the existence of a Rab8-dependent basal route. Third, our data confirm the existence of a cryptic route towards the apical surface via Rab11 [23]. Thus, these three different Rab proteins, Rab11, Rab10 and

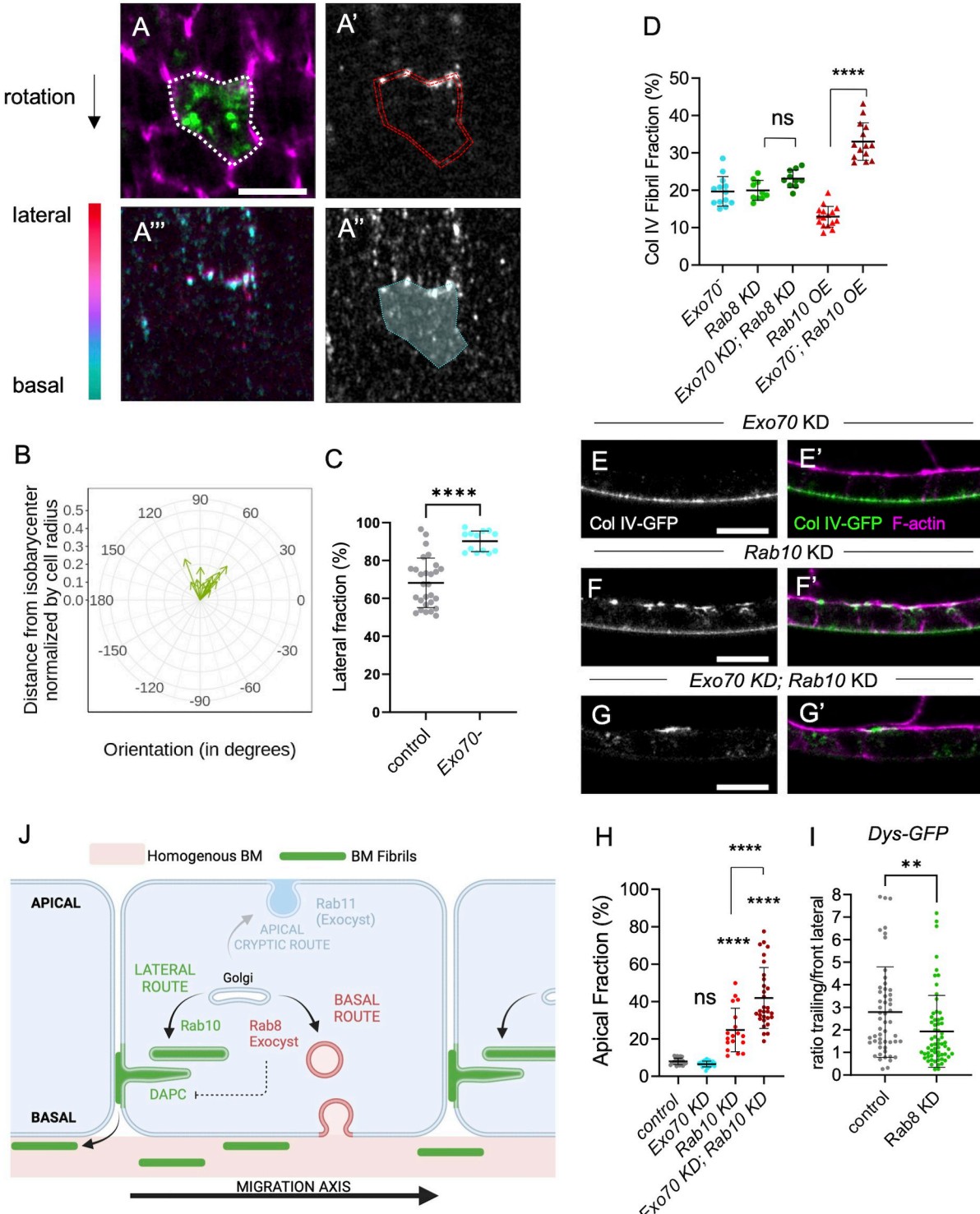

**Fig 7. *Exocyst is required for the Rab8 dependent route*.** (A) Lateral (A-A') and basal (A") focal planes or color-coded z projection in function of the z axis of the 12 lateral and basal focal planes used for BM secretion quantification (A"') in *Exo70* null mutant individual single-cell clones that express Col IV-GFP (green, top), stained for F-actin (magenta) and immunostained for GFP without permeabilization (white). Scale bars, 5 µm. Dotty white line (A) highlights the cell periphery. Double dotty red lines (A') and white surface delimited by dotty cyan line (A") indicate the surface of the given z-section taken into account for lateral or basal secretion quantification respectively. Scale bars, 5 µm. (B) Quantification of the planar orientation of the lateral BM protein secretion in *Exo70* null mutant single cells. (C) Quantification of the lateral fraction of BM secretion (%) in control and *Exo70* null mutant single-cell clones. In order on the graph, n = 28 and 13 cells. (D) Quantification

of the BM fibril fraction (%) in stage 8 follicles of the indicated genotypes. In order on the graph, n = 10, 10, 13, 16, 14 follicles. (E-G) Cross-sections of stage 8 follicles of the indicated genotypes: (E, E') *tj> Exo70 RNAi*, (F, F') *tj> Rab10 RNAi*, (G, G') *tj > Exo70 RNAi > Rab10 RNAi*. Images show Col IV-GFP (white, left; green, right) and F-actin (magenta) localization. Scale bar, 10 μm. (H) Quantification of the ratio between apical and basal Col IV-GFP fluorescence intensity (%) in stage 8 follicles of the indicated genotypes. In order on the graph, n = 24, 23, 18, 30 follicles. (I) Quantification of Dys-GFP mean signal intensity at the trailing edge vs adjacent lateral front side in control and *Rab8* KD follicle cells. In order on the graph, n = 53 and 63 follicle cells. J) Scheme representing the three different BM secretion routes with their cell exit site and their components. For all graphs, data are the mean ± SD; ns, not significant, *p <0.05, ****p <0.0001 (Unpaired t test for C, Ordinary one-way ANOVA with Tukey's multiple comparisons test for D, H; Mann-Whitney test for I).

Rab8, are targeted towards the three distinct domains of epithelial cells defined by apical basal polarity, apical, lateral and basal, respectively. Moreover, Rab10 is also planar polarized. The same three Rab GTPases have been jointly involved in different processes such as ciliogenesis, targeted exocytosis or lysosome homeostasis where they have been proposed to act in a redundant manner [43–47]. However, these observations mainly came from mammalian cell culture systems with unpolarized cells or from pathological situations where apical basal polarity was disrupted. Here, these routes could be considered as partially redundant because they are all involved in BM protein secretion and all three must be knocked down to completely block it. However, the integration of these routes to cell polarity clearly allowed distinguishing the spatial specificity of these three Rab proteins and revealed that they compete with each other.

The next obvious question is how these routes are differentially integrated with cell polarity. Part of the answer likely comes from microtubule-associated transport, and two kinesins, Khc-73 and Khc, involved in the basolateral targeting of BM-containing vesicles [28]. Their concomitant loss of function phenocopies, to some extent, the double *Rab8-Rab10* KD, suggesting that both routes of BM protein basolateral secretion are affected when these kinesins are absent. While a direct link between Rab10 and the Khc-73 homolog has been established, one could speculate that Kin1 activity might be more related to the Rab8 route [32]. Nonetheless, our data indicate that microtubule transport is not the only difference between these routes.

We established a functional link between the exocyst complex and Rab8. This link has been shown in other systems, including BM remodeling, suggesting that it might be a general feature of the Rab8-dependent route [48–52]. However, we showed, for the first time, the involvement of exocyst in BM secretion. Importantly, the absence of basal secretion in *Exo70* mutants and the genetic evidence of Exo70 requirement for the Rab8 route confirm that the Rab8 route is targeted towards the basal domain of the cells.

Moreover, several of our results are in favor of a direct involvement of Dys and Dg in the Rab10 route. First, Dys is highly associated with the Rab10 compartment, and DAPC is required for BM fibril formation. Second, in the absence of Dys, BM protein diversion towards the apical domain is increased upon Rab8 KD. This result excludes a model in which Dys participates in BM fibril formation only after BM protein secretion. Moreover, we showed that Dg distribution all around the cells, when overexpressed, is sufficient to redistribute Rab10 and to induce aberrant BM secretion, confirming a functional link. However, we could not detect any alteration in Rab10 localization and collagen lateral secretion in the absence of Dys. Therefore, more investigations are required to define exactly how the DAPC allows the formation of BM fibrils. Nonetheless, given the importance of the DAPC and BM proteins in muscular dystrophies, our results will pave the way to determine whether a similar function is present also in muscle cells [53]. Interestingly, the extracellular matrix is different between the myotendinous junction and the interjunctional sarcolemmal basement membrane and may provide another developmental context where several routes targeted to different subcellular domains may be implicated [54].

Lastly, we revealed another level of regulation of this process based on the direct interaction between Dys and Exo70 to modulate Dys localization. Functional links have been proposed between the exocyst complex and Dg, but they had never been associated through a direct

molecular link and Dys involvement was not shown [38,39]. The high conservation of the interaction domain on Dys, a domain with no other known interacting protein, suggests that this interaction has been conserved during evolution and may be relevant in other contexts.

Our work identified distinct actors of the Rab8- and Rab10-dependent routes for BM protein secretion, with the involvement of the DAPC and exocyst complexes. However, more studies are required to understand the underpinnings of such specificity between these very close Rab GTPases. The fact that Rab8 and Rab10 share a similar function (BM protein secretion), but with a different output (homogenous vs fibrillar BM), in a highly tractable system for genetics and imaging, offers a unique opportunity to thoroughly investigate their molecular and mechanistic specificities. On the basis of the present results, we propose the co-existence of parallel secretory routes orchestrated by these Rab proteins with different cell exit sites defined by cell polarity axes as the basic mechanism to explain BM diversification during development.

## Material and methods

### Genetics

All the fly strains used are described in S1 Table The detailed final genotypes, temperature and heat-shock conditions are given in S2 Table.

### Dissection and immunostaining

Resources and reagents are listed in S1 Table. Ovaries were dissected in supplemented Schneider medium, ovarioles were separated and the muscle sheath was removed before fixation to obtain undistorted follicles. Ovaries were fixed with 4% paraformaldehyde in PBS containing phalloidin Atto-488, phalloidin Atto-550 or phalloidin Atto-633 (Sigma) for 20 min, washed 3 times in PBS, and permeabilized with 0.02% Triton X-100 containing phalloidin Atto-488, -550 or -633 to stain F-actin. For antibody staining, dissected ovaries were fixed for 15–20 min with 4% formaldehyde in PBS containing phalloidin Atto-488, phalloidin Atto-550 or phalloidin Atto-633 (Sigma) and addition of 1 mM CaCl2 when anti-E-Cad antibody was used. After blocking and permeabilization with PBS, 0.5% BSA and 0.2% Triton X-100 (PBT), primary antibodies were diluted in PBT and detected with Alexa Fluor-conjugated secondary antibodies. Antibodies are described in S1 Table. F-actin was labeled with phalloidin Atto-488, -550 or -633. For non-permeabilized experiments, the anti-GFP antibody to detect pericellular BM was used as described above using PBS, 0.5% BSA (PBS-BSA) instead of PBT. For collagenase treatment, dissected ovarioles were incubated in 100μl supplemented Schneider medium containing 165 U of collagenase for 10 min followed by three washes in supplemented Schneider medium before fixation with 4% paraformaldehyde in PBS, phalloidin Atto-633, for 20 min. Images were taken using a Leica SP8 (for fibril fraction determination) or a Zeiss LSM800 Airyscan or a Zeiss LSM960 Airyscan2 confocal microscope.

### Cloning and transgenesis

Scarlet and Exo70 cDNAs were cloned in frame in pUASz using the EBuilder HiFi DNA Assembly Cloning Kit. Site-directed transgenesis was performed at the attP40 site. The sequence corresponding to the Dys spectrin repeats 22–23 (amino acids 2549–2636 of the DysPH isoform) were cloned in frame in the pGex4T3.1. The Dys::sfGFP knock-in line was generated by the inDroso Functional Genomics (Rennes, France) using the following strategy. After cleavage with a gRNA (GACGACGACCACGGCAACCA CGG), a cassette that contained the Dys coding sequence end in frame with sfGFP and a selection marker between the

LoxP sites was introduced by recombination. After obtaining the transgenic line, the selection marker was removed by crosses with a CreLox line.

## GST pull-down experiments

GST and GST-DysSR22-23 were produced in BL21 bacteria using a standard protocol and were purified on Glutathione Sepharose 4B (GE Healthcare) [55]. Exo70 was produced by *in vitro* transcription and translation in the presence of $S^{35}$ using standard protocols. Interactions were tested in 50mM Tris HCl, 100mM NaCl, 0.01%Triton-X100, 0.1% BSA at 20˚C for 12 hours before washes in 50mM Tris HCl, 100mM NaCl, 0.01%Triton.

## BM fibril fraction, BM apical fraction and basal BM intensity

BM fibril fraction (FF) was determined using a homemade Fiji macro, as previously described [26]. The BM apical fraction was quantified using the Fiji tool to measure and compare the 'mean gray value' of pairs of 6 pixels (1.14 μm) thick regions of interest (ROIs) drawn in the equatorial region of sagittal view of ovarian follicles, alongside the cell apical and basal walls (5 to 15 cells per ROI, 2–4 pairs of ROIs per follicle, about 25 pairs of ROIs for each genotype). Basal BM intensity was determined by calculating the 'mean gray value'/$μm^2$ in the ROI drawn at the basal wall.

## Dg, Dys and Rab10 trailing vs front lateral signal intensity ratio

The Fiji segmented line tool was used to manually trace two lines per follicle cell, following the F-actin signal: one line on the trailing edge and the other one along one of the two front lateral sides of the cell, on a projection of three focal plans with a Z step of 0.15 μm starting at ~0.15 μm away from the basal surface. A third line, drawn away from the F-actin signal, was used for background subtraction. The 'mean gray values' for Dg, Dys, and Rab10 signal were determined and the trailing / front lateral signal ratio was plotted.

## BM protein secretion quantification

Follicle cells were manually segmented based on F-actin signal, on nine successive focal plans with a Z step of 0.15 μm, from the basal surface (detected as the Z focal plan giving the best signal for actin stress fibers) towards the apical surface. BM protein secretion was quantified using a homemade Fiji macro to detect the signal intensity given by anti-GFP antibody immunostaining without permeabilization in single-cell clones that express UAS:Col IV-GFP. The lateral secretion orientation was assessed by determining the position of the barycenter of the hollow cylinder of 1.5 μm thickness to model the nine ROIs. The signal intensity and direction, defined by the vector between the isobarycenter of the cylinder and its center of mass, was measured using the signal intensity as weight. The basal secretion was assessed by measuring the signal intensity of a full cylinder made by projecting the ROI drawn on the basal surface, and shrunk to 0.75 μm in diameter, on the three slices beyond the cell surface. The fraction of laterally secreted BM proteins was determined as the percentage of signal intensity from the lateral secretion relative to the total signal intensity measured from the two cylinders. A ROI drawn on non-GFP cells and reported on the same nine lateral Z focal plans and three slices beyond basal surface was used to subtract the background lateral and basal signals.

## Statistical analyses

For all experiments, the minimum sample size is indicated in the figure legends. Results were obtained from at least two independent experiments, and for each experiment multiple females

were dissected. Randomization or blinding was not performed. Data normality was calculated using the D'Agostino and Pearson normality test. The unpaired t-test and ordinary one-way Anova were used to compare respectively 2 or more samples with normal distribution, and the Mann-Whitney and Kruskal-Wallis tests were used to compare respectively 2 or more samples without normal distribution. Graphs were generated with Prism software. All numerical data are given in S3 Table.

## Supporting information

**S1 Fig.** (A) Cross-sections of stage 8 ovarian follicles showing Col IV-GFP (green, top; white, bottom) and F-actin (magenta) localization in control line (left) or *tj>Rab10* RNAi *(*Rab10 knock-down) indicated lines: JF02058, KK109210 or GD13414. (B) Apical fraction quantification (%) of Col IV-GFP fluorescence intensity in stage 8 follicles of the indicated genotypes. Data are the mean ± SD; ****p <0.0001 (Ordinary one-way ANOVA with Tukey's comparison test); n = 14, 14, 13, 14 follicles. (C) Projections of the apical region of stage 8 follicles that capture some of the apical surface due to the tissue curvature showing the ectopic Col IV-GFP (white, top; green, bottom) and F-actin (white, middle; magenta, bottom) in follicular epithelia of the indicated genotypes: control, *Rab8* KD, *Rab10* KD and *Rab8-Rab10* double KD. (D) Cross-sections of stage 8 follicles of the indicated genotypes (control, *Rab8-Rab10* double KD, *Rab11* KD, *Rab8-Rab10-Rab11* triple KD, showing aPKC (magenta, 1st line; white, 2nd line), Dlg (yellow, 3rd line; white, 4th line), E-Cad (orange, 5th line; white, 6th line) and Col IV-GFP (green).
(TIFF)

**S2 Fig. Basal view of the BM at stage 8 in *Col IV-GFP* and *tj> Cg25c-GFP* follicle cells.** Quantifications of BM fibril fraction (%) in stage 8 follicles of the indicated genotypes. Data are the mean ± SD; ns, not significant (unpaired t test). In order on the graph, n = 10, 9 follicles.
(TIFF)

**S3 Fig.** (A-C') Basal (C, C'), suprabasal (i.e. 0.4 μm above the basal surface) (B, B') and middle (i.e. 0.8 μm below basal surface) (A, A') views of stage 8 follicles that overexpress RFP-tagged *Rab10* (magenta) and endogenous Col IV-GFP (green). (D-D''') Middle view of stage 8 follicles that overexpress RFP-tagged *Rab10* (black in 1st and 3rd line, white in 4th line), express endogenous Col IV-GFP (green) and that are stained with anti-golgin-245 antibody (magenta) and F-actin (white in 2nd line). (E-E'') Basal view of stage 8 follicles treated with collagenase before fixation to visualize intracellular endogenous basal Col IV-GFP (green) and Rab10-RFP (magenta). (F-F'') Basal view of a photobleached window at the surface of stage 8 living follicles taken 20 minutes after photobleaching to visualize intracellular basal Rab10-RFP (magenta) and endogenous Col IV-GFP (green) expression in non-bleached cells that have migrated in the bleached BM area. Scale bars, 5 μm.
(TIFF)

**S4 Fig.** (A-A') Basal and (A'',A''') suprabasal view of stage 8 follicles expressing *Rab10*-RFP (magenta) in WT cells (GFP cells, green) or in *Dys* null mutant cell clones (no GFP). (B) Rab10-RFP mean signal intensity at the trailing edge vs adjacent lateral front side of follicle cells of the indicated genotypes, on 0.45 μm projections starting 0.15 μm below the basal surface. Data are the mean ± SD. In order on graph, n = 73 and 56 follicle cells. (C) Quantification of the lateral fraction of BM secretion (%) in single-cell clones overexpressing Col IV-GFP in control and *Dys* null mutant follicles. Data are the mean ± SD. In order on graph, n = 21 and 18 cells. ns, not significant (unpaired t test). (D) Quantification of the planar orientation of lateral BM protein secretion from individual single-cell clones in control (red) and *Dys* null

mutant (blue). (E-G) Sagittal view of follicles overexpressing *Dg*-GFP (E, E') or expressing endogenous *Dys*-sfGFP in WT context (F, F') or in *Dg* overexpression (G-G'). *Dg*-GFP is shown in white, *Dys*-sfGFP in green and F-actin in magenta. (H, I) basal and (H', I') suprabasal images of tj>Dg-GFP in (H) WT or (I) *Dys* mutant conditions. (J) Quantifications of BM fibril fraction (%) of stage 8 WT or *tj> Dg* follicles. Data represent mean ± SD. Unpaired t test, *p < 0.05. In order on graph, n = 14 and 11 follicles. (K-L) Basal and (K', L') suprabasal views of the BM on stage 8 control (K, K') and *tj> Dg* (L, L')) follicles visualized with Col IV-GFP (green). Scale bars, 5 μm in (A, H-I'), 10μm in (E-G' and K-L')
(TIFF)

**S5 Fig.** (A) Quantifications of BM fibril fraction (%) in stage 8 follicles of the indicated genotypes. In order on the graph, n = 12, 9, 8, 9, 12, 9, 9, 12 and 10 follicles. (B-D) Basal view of the BM visualized with Col IV-GFP at stage 8 in (B) control, (C) *Exo70* null mutant, (D) *Exo70* null mutant with *Exo70 OE* follicles. Scale bars, 10 μm. (E) Quantification of BM fibril fraction (%) in stage 8 follicles of the indicated genotypes. In order on the graph, n = 13, 11, 11 follicles. For all graphs, data are the mean ± SD; ***p <0.01, ****p <0.0001 (Ordinary one-way ANOVA with Dunnett's multiple comparisons test in A, and Kruskal-Wallis with Dunn's comparisons test in E).
(TIFF)

**S1 Table. Resources and reagents.**
(DOCX)

**S2 Table. Genotypes and specific conditions.** (h: hours; HS: heat-shock).
(DOCX)

**S3 Table. Numerical data and statistical analysis supporting figures.**
(XLSX)

## Acknowledgments

We are grateful to M.-L. Parmentier and V. Van De Bor for sharing fly stocks. We also thank the CLIC facility (Clermont Imagerie Confocale).

## Author Contributions

**Conceptualization:** Cynthia Dennis, Vincent Mirouse.

**Formal analysis:** Cynthia Dennis, Vincent Mirouse.

**Funding acquisition:** Vincent Mirouse.

**Investigation:** Cynthia Dennis, Vincent Mirouse.

**Methodology:** Cynthia Dennis, Pierre Pouchin, Graziella Richard.

**Project administration:** Vincent Mirouse.

**Software:** Pierre Pouchin.

**Supervision:** Vincent Mirouse.

**Writing – original draft:** Cynthia Dennis, Vincent Mirouse.

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
