## [Decision Letter · Decision Letter 0]

5 Feb 2024

Dear Dr Mirouse,

We are pleased to inform you that your manuscript entitled "Basement membrane diversification relies on two competitive secretory routes defined by Rab10 and Rab8 and modulated by dystrophin and the exocyst complex" has been editorially accepted for publication in PLOS Genetics. Congratulations!

The manuscript was evaluated by reviewer #2 of the original review commons submission; as you will see, they recommend acceptance.

Yours sincerely,

Gregory S. Barsh

Editor-in-Chief

PLOS Genetics

Gregory Copenhaver

Editor-in-Chief

PLOS Genetics

Comments from the reviewers (if applicable):

Reviewer's Responses to Questions

**Comments to the Authors:**

Reviewer #1: Reviewer 2 thinks that the authors appropriately answered all their comments raised in the first round of revision. The modifications made by the authors throughout the manuscript significantly improve their manuscript. The revaluation of their data through alternative quantification, and the addition of new schematics strengthen their manuscript.

**Have all data underlying the figures and results presented in the manuscript been provided?**

Reviewer #1: Yes

PLOS authors have the option to publish the peer review history of their article (what does this mean?). If published, this will include your full peer review and any attached files.

Reviewer #1: No

**Data Deposition**

http://datadryad.org/submit?journalID=pgenetics&manu=PGENETICS-D-23-01417

**Press Queries**

---

## [Editor Report · Acceptance letter]

28 Feb 2024

PGENETICS-D-23-01417 

Basement membrane diversification relies on two competitive secretory routes defined by Rab10 and Rab8 and modulated by dystrophin and the exocyst complex 

Dear Dr Mirouse, 

We are pleased to inform you that your manuscript entitled "Basement membrane diversification relies on two competitive secretory routes defined by Rab10 and Rab8 and modulated by dystrophin and the exocyst complex" has been formally accepted for publication in PLOS Genetics! Your manuscript is now with our production department and you will be notified of the publication date in due course.

With kind regards,

Judit Kozma

PLOS Genetics

On behalf of:
